# Sexual and Gender Minorities and Risk Behaviours among University Students in Italy

**DOI:** 10.3390/ijerph182111724

**Published:** 2021-11-08

**Authors:** Marco Fonzo, Silvia Cocchio, Matteo Centomo, Tatjana Baldovin, Alessandra Buja, Silvia Majori, Vincenzo Baldo, Chiara Bertoncello

**Affiliations:** 1Hygiene and Public Health Unit, Department of Cardiac Thoracic and Vascular Sciences and Public Health, University of Padova, 35131 Padova, Italy; silvia.cocchio@unipd.it (S.C.); matteo.centomo@studenti.unipd.it (M.C.); tatjana.baldovin@unipd.it (T.B.); alessandra.buja@unipd.it (A.B.); vincenzo.baldo@unipd.it (V.B.); chiara.bertoncello@unipd.it (C.B.); 2Department of Public Health and Community Medicine, Hygiene and Environmental, Occupational and Preventive Medicine Division, University of Verona, 37129 Verona, Italy; silvia.majori@univr.it

**Keywords:** sexual and gender minorities, LGBTQ+, behavioural risk factors, young adults

## Abstract

Sexual and gender minorities (SGM) may experience stigma, discrimination and show higher prevalence of behavioural risk factors than heterosexual counterparts. In Italy, the information on SGM is scarce and outdated. The present cross-sectional study aims to provide a more up-to-date estimate of the SGM proportion in young adults and to assess differences in the adoption of risk behaviours compared to their heterosexual counterparts. The study involved university students aged 18–25. Information on socio-demographic and behavioural characteristics were collected. The effect of sexual orientation on risk behaviours was assessed with a multinomial single-step logistic regression analysis. A total of 9988 participants were included. Overall, 518 students (5.2%) self-identified as SGM. While lesbians showed significantly higher odds of only non-regular use of protective barriers (AOR: 11.16), bisexuals showed higher odds for frequent drinking (AOR: 2.67), smoking (AOR: 1.85), multiple sexual partnerships (AOR: 1.78) and non-regular use of protective barriers (AOR: 1.90) compared with heterosexual women. Gay men showed higher odds of multiple sexual partnerships compared with heterosexual males (AOR: 5.52). SGM accounted for 5.2% of the sample, slightly more than the proportion found in the general population, but substantially in line with similarly aged populations abroad. Our findings confirm that unhealthy risk behaviours are more frequent among LGBTQ+, in particular among bisexual women. Rather than targeting specific subpopulations, our study aims to show the need for health promotion interventions that aim at the empowerment of all students regardless of sexual orientation, being aware that SGMs can benefit to a relatively greater extent.

## 1. Introduction

Recent literature estimates that the proportion of sexual and gender minorities (SGM) stands at around 3.8% in the general population [1]. In Italy, the latest available data (census 2011) show that 2.4% of the general population self-identify as SGM, with a higher proportion among young adults [2]. Similarly, around 3.5% of adults in the United States identified as MSM/WSW or bisexual, with a slightly higher proportion among males (3.6%) than among females (3.4%) [1,3]. Several studies are consistent in reporting that the proportion of bisexuals is higher than WSW among females, while quite the opposite is the case among males [4,5,6,7]. Available data on transgender individuals are fewer. Current estimates indicate a proportion ranging between 0.1 and 0.5% [3,8,9].

However, the proportion of SGM is not immutable in time and space. The age and the education of the interviewed population, the place, the cultural context and even the year in which the survey was conducted should be considered. Changes, although seemingly secular, could take place in a relatively short period of time [2,4,10,11,12,13,14].

SGM often experience stigma and discrimination, conditions associated with psychological distress that frequently result in health risk behaviours [6]. Several studies have confirmed the existence of psychological symptoms in the LGBTQ+ population, such as anxiety and stress, higher prevalence of health risk behaviours and vulnerability for physical and mental health [15]. A recent review found that among SGM, the risk of suffering from depression or anxiety disorders was 1.5 times higher than among heterosexual people [16]. Other studies found that gay and bisexual men have higher levels of depression, panic attacks and psychological distress and that lesbian and bisexual women show a higher prevalence of generalised anxiety [17]. LGBTQ+ adolescents also report higher levels of depressive and anxiety symptoms than their heterosexual peers [18]. In addition to anxiety and depressive symptoms, suicidal ideation, alcohol abuse, problems in the workplace, academics, leisure time and interpersonal relationships have been found. Previous research showed that 11.6% of gay men and 10.6% of lesbian women experience social-emotional symptoms that cross the borderline of social distress in terms of both anxiety and interpersonal and social relationships [15]. Thus, one can speak of minority stress to indicate the excess stress to which individuals from social categories that are stigmatised because of their social position, often a minority one, are exposed. That is, there is evidence of the negative effects of social conditions, such as prejudice and stigma, on the lives of individuals and minority groups [19,20]. Overall, growing evidence shows how alcohol consumption [11,21,22,23] and smoking [5,24] are higher in SGM, as well as the average number of sexual partners [25,26].

In Italy, the information on SGM is scarce and latest demographic data in this regard date back to the 2011 census. In addition, the prevalence of risk behaviours in LGBTQ+ young adults has not been previously estimated. The present study aims to provide a more up-to-date estimate of the proportion of SGM in young adults and to assess the probability of the adoption of health risk behaviours in SGM compared to their heterosexual counterparts.

## 2. Materials and Methods

The study was conducted from October 2015 to June 2016 at the University of Padova and the University of Verona, Italy. Participants were eligible if they were 18–25 years old and enrolled in one of the two universities. The participation was on a completely free and voluntary basis and no form of compensation was offered. Members of the research group visited the sampled classrooms and all students in attendance were invited to complete the questionnaire. Participants were given oral information about the purpose of the study, and reassurance on its confidentiality and their anonymity. A written questionnaire was administered to each participant.

The questionnaire investigated the following socio-demographic and behavioural factors: age (entered as a continuous variable); nationality (citizenship); degree course; gender identity; sexual orientation (heterosexual, bisexual, gay/lesbian); involvement in a relationship; alcohol consumption; smoking habit; number of sexual partners within the previous 24 months; age at the sexual onset; use of condoms (or other protective barriers). Degree courses attended by participants were classified as follows: (a) healthcare professions, (b) sciences, (c) humanities, (d) engineering, (e) law and economics, (f) psychology, (g) medicine and dentistry. Participants with an alcohol consumption of 6 or more days a week were coded as ‘frequent drinker’. In the subsequent analyses, a sexual onset at the age of 14 or earlier was considered as ‘early sexual onset’ and a number of three or more sexual partners in the previous 24 months was coded as ‘multiple sexual partnership’. Pretesting of the questionnaire was conducted; findings from the pilot study and subsequent changes were described in detail elsewhere [27].

Descriptive analyses, using absolute and relative frequencies, were performed. Chi-squared tests were conducted to compare the different groups on baseline characteristics. The effect of sexual orientation on the adoption of risk behaviours was assessed with a multinomial single-step logistic regression analysis adjusting for age, nationality, degree course, involvement in a relationship, alcohol consumption, smoking habit, number of sexual partners within the previous 24 months, sexual onset, and use of condoms (or other protective barriers). Statistical significance for all tests was set at *p* < 0.05 (two-sided) and confidence intervals (CIs) were set at 95%. Statistical analyses were performed using IBM^®^ SPSS Statistics^®^ version 23. Items and data in this study were collected, analysed and presented according to criteria from the STROBE Statement for cross-sectional studies. Written informed consent was obtained and collected separately from the questionnaire in order to ensure the anonymity of the data. The study protocol was approved by the ethics committee of the Padua Provincial Authority on 30 July 2015 (ID: 3404/U16/15) [28].

## 3. Results

Out of 11,096 candidates, 10,135 students agreed to be included in the study (response rate: 91.3%). After checking for completeness and failure to adhere to inclusion criteria, 9988 questionnaires were included in the study (90.0% over the initial quota of the candidates).

The mean age of participants was 20.5 years (SD: 1.81), and the median age was 20.0 years. Participants with Italian citizenship comprised 97.4% of the sample. Students committed to a stable relationship constituted 44.8% of the total participants. Overall, 518 students (5.2%) self-identified as SGM. Among cisgender males, 2.59% declared themselves as MSM, while 2.02% declared themselves as bisexual. Among cisgender females, 1.28% declared as WSW, while 4.21% declared as bisexual. Transgender individuals accounted for 0.12% of the sample. Fully detailed socio-demographic characteristics of the study population are shown in Table 1.

While no statistical differences in the pattern of alcohol consumption were found among males, a higher proportion of bisexual women used to drink alcohol more than 6 days a week (5.4% compared with 1.3 and 0.0% of heterosexual women and lesbians, respectively), or 2–5 days a week (22.9% compared with 14.1 and 11.6% of heterosexual women and lesbians, respectively), as shown in Table 2.

In a similar manner, while among males, smoking habits did not show significant differences among the three subgroups, the proportion of non-smokers was lower in bisexuals (48.4%) and WSW (59.5%) compared with heterosexual women (73.9%). Speaking of the number of sexual partners, the proportion of bisexual women that had three or more partners in the past two years (26.3%) was more than twice that of heterosexuals (11.1%) and lesbians (12.5%); among males, the proportion was 18.7% among heterosexuals, 32.4% among bisexuals, and 49.4% among MSM. An early sexual onset was reported by 12.0% of bisexual women, while the proportion was lower in heterosexuals and lesbians (5.8 and 3.6%, respectively). Among gay men, the proportion of those who had an early sexual onset was slightly higher than their heterosexual counterparts (5.9 vs. 4.9%), while the proportion of those who declared a sexual onset after the age of 19 was almost double (31.8 vs. 16.0%). No significant differences were noted regarding the use of condoms among males, while the proportion of WSW who declared regular use of condoms (19.0%) was lower than both hetero- and bisexuals (55.5 and 43.4%, respectively).

Results from the logistic regression are reported in Table 3. While lesbians showed significantly higher odds of non-regular use of condoms only (AOR: 11.16; 95%CI: 4.78–26.05), bisexuals showed higher odds for all risk factors investigated except for early sexual onset: frequent drinking (AOR: 2.67; 95%CI: 1.33–5.36), smoking (AOR: 1.85; 95CI; 1.33–2.58), multiple sexual partnerships (AOR: 1.78; 95%CI: 1.21–2.61) and non-regular use of condoms (AOR: 1.90; 95%CI: 1.3–2.70). Among males, MSM showed significantly higher odds of multiple sexual partnerships compared with their heterosexual counterparts (AOR: 5.52; 95%CI: 3.26–9.35).

## 4. Discussion

Overall, SGM accounted for 5.2% of the study population. Considering the data from the latest national census in 2011, which reported a proportion of 3.2% in the age range of 18–25, our findings show a higher proportion of SGM [2]. However, the comparison must be conducted with caution: in the 2011 census, 15.6% of participants did not answer this specific question and 5.0% identified neither as heterosexual nor as LGBTQ+, and thus, the proportion was probably underestimated; in addition, in the general population, the percentage of LGBTQ+ responders was higher in the North of Italy (3.1%) compared with the Centre (2.1%) and the South (1.6%). Even considering this further specification, our study (which was conducted in Northeast Italy) showed a higher proportion of SGM people. The university student population may not be representative of the general population of same age because of a higher educational level and a different cultural context. Indeed, studies from the United States conducted on university students in the 2000s show how LGBTQ+ individuals accounted for around 5%; thus, our sample does not seem to be an exception compared with other comparable contexts [14,29]. Our study also confirms a higher proportion of bisexual individuals among females compared with males (4.2 vs. 2.0%), while the proportion of homosexuals was higher in males (2.6%) compared to females (1.3%), as established in previous studies [5,6,7]. Transgender individuals accounted for 0.1% of the study population, in line with available data from the United States [3] and the United Kingdom [9].

Regarding the prevalence of risk behaviours, the main finding of our study is that while, among males, no significant differences were noted between SGM and their heterosexual counterparts (with the exception of a higher number of sexual partners among gay men), among women, bisexuality was significantly associated with frequent alcohol drinking and smoking, multiple sexual partnerships, and non-regular use of condoms or other protective barriers; in contrast, lesbians did not show higher odds for any of the risk factors investigated compared to heterosexual females, with the exception of a non-regular use of condoms or other protective barriers.

Most of the students in the sample used to drink alcohol once a week or less and were non-smokers regardless their gender identity or sexual orientation. However, while, among males, there were no significant differences, bisexual women reported a significantly higher alcohol consumption compared with heterosexual females and lesbians, and their odds were more than twice as high (AOR: 2.67; 95%CI: 1.33–5.36); almost the same applied to smoking (AOR: 1.85; 95%CI: 1.33–2.58). In our sample, gay and bisexual men did not show higher odds of smoking compared with heterosexual males. This leaves the debate open, as there are studies reporting a similar result [13,30] but also others showing a higher prevalence of smokers in SGM males [31,32,33]. Our findings partially challenge the previous evidence showing that LGBTQ+ as a whole are at higher risk for alcohol and tobacco use [11,21,22,23]. It is essential to make a distinction: the highest risks are for bisexual women. A possible explanation could be that although it is well known that SGM may be subject to an excess of stress due to social stigma, prejudice and the marginalizing nature of social environments—better defined as minority stress [34]—not all subgroups may be exposed in the same way. Although many studies showed that bisexuals seem to be more exposed [5,11,13,21,23,24], the root causes are not yet fully understood. As a hypothesis, bisexual women may not have the opportunity to feel a sense of belonging to a protective social network based on their sexual orientation, as may be the case for WSW/MSM. In support of this, some studies demonstrated that the social network surrounding bisexual individuals may be more scarce since bisexual adults showed the lowest life satisfaction on average and reported that they received the least emotional support, compared with their hetero- and homosexual peers [4,13,35]. Bisexual women may, therefore, experience greater social exclusion and ‘twice’ the minority stress: a sort of minority within the minority.

Multiple sexual partnerships were reported by 1 in 4 (26.3%) bisexual women and almost 1 in 2 gay men (49.2%), with an AOR of 1.78 (95%CI: 1.21–2.61) and 5.52 (95%CI: 3.26–9.35), respectively, compared with their heterosexual counterparts. Again, our findings partially challenge previous results by dissecting the SGM into subgroups: thus, although we confirm the tendency to have multiple sexual partners in gay men and bisexual females, our results align lesbian women and bisexual men to their heterosexual counterparts [26].

With regard to the use of condoms or other protective barriers in SGM, an interesting finding comes from a comparison with an US national survey on college students by Eisenberg and colleagues: in our sample, bisexuality among females was associated with a non-regular use of condoms (AOR: 1.90; 95%CI: 1.33–2.70), while no significant difference was noted in the US survey; conversely, in the US survey, MSM showed significantly higher odds for non-regular use of condoms (the AOR for always using a condom was 0.61; 95%CI: 0.39–0.96), while, in our sample, gay men did not show a significant difference with heterosexual males in terms of the use of condoms [36]. Moreover, males in our sample showed a higher use of condoms compared with the US, especially as regards gays (71.1 vs. 35.5%). The US survey was conducted in the mid-1990s, which was around 20 years before our study. However, the lag time may not be the only factor responsible for this result. The proportion of gay men always using a condom in our sample was, in any case, higher than other studies conducted in young adults globally [25,37,38,39]. This finding is definitely encouraging, and it may be a positive consequence of more than thirty years of efforts in public health against the spread of HIV infection with a specific focus on gay men, historically targeted as a population at high risk. Unfortunately, Eisenberg and colleagues did not show data on the use of condoms or other protective barriers in lesbians. In our sample, WSW showed significantly higher odds of non-use of condoms (AOR: 11.16; 95%CI: 4.78–26.05) with only 19.0% of them claiming to use a condom or other protective barriers. This datum does not deviate substantially from that shown in a survey conducted in the US in 2008–2009 considering female college undergraduate students (in this latter case, the use of condoms or other protective barriers stood at 16.3% in lesbians compared with 60% of heterosexual and 50.5% of bisexual females) [40]. As suggested by Oswalt and Wyatt, this gap may be due to a prevailing focus of safer sex campaigns on the risks connected with sexual intercourse [41], thus leaving aside other sexual practices, both penetrative and non-penetrative, that may have a different diffusion in SGM, or due to a communication strategy that has been focusing only on the use of protective barriers as contraceptive methods. However, the perception that sexually transmitted infections should only be a problem for heterosexual and bisexual women still holds sway. [42].

Some participants may have had concerns in self-identifying as non-heterosexuals and may have provided answers perceived as being correct according to social norms. A social desirability bias may more easily affect face-to-face interviews; however, also in self-administered anonymous questionnaires, the presence of a social desirability bias cannot be excluded [43]. In addition, results on risk behaviours may not be inferred to the whole Italian population in the same age range because the study population consisted of only university students. Given the study protocol, the study design and permissions, it was not possible to retrieve any information on students who did not agree to participate. Although the definitions are complex and there is no unanimous consensus, for the purpose of this study, we felt it was legitimate to use the terms ‘Sexual and Gender Minorities’ and ‘LGBTQ+’ as synonyms in the current document, as well as the terms lesbian/WSW and gay/MSM for female and male homosexuals, respectively. Despite the abovementioned limitations, a strength that is worth mentioning regarding this study comes from the involvement of a very large sample size of around 10,000 participants.

## 5. Conclusions

In conclusion, the proportion of SGM among young adults in our sample stood at around 5.2%, substantially higher than in the Italian general population, but in line with similar-aged populations, as reported in studies conducted abroad. Italy does not seem to be an exception in the proportion of gay/lesbian and bisexual individuals in females (1.3 and 4.2%, respectively) and males (2.6 and 2.0%, respectively), as well as the proportion of transgender individuals (0.1%). Compared with their heterosexual counterparts, SGM individuals showed higher odds of risk behaviours but the burden was not equally distributed within the SGM: whereas, in males, no overall significant differences were noted according to sexual orientation (except for gay men whose odds of having multiple sexual partnerships were more than five-times higher than heterosexuals), bisexuality in females was associated with a higher risk for frequent alcohol drinking, smoking, multiple sexual partnerships and non-regular use of condoms or other protective barriers. Compared to the past, the regular use of condoms in gay men is particularly high.

Our research shows that SGM university students are at higher risk of adopting unhealthy behaviours in Italy and supports the use of easy-to-implement and proven effective public health interventions to actively increase the empowerment of university students such as short seminars, short counselling activities and well-planned communication campaigns. For example, studies on raising awareness of condom use among university women have shown the importance of conveying information appropriately and one of the most efficient ways is through health education websites that are certified and reliable, such as the university website; similarly, short online seminars or self-assessment questionnaires followed by a tailored risk assessment feedback have been shown to be effective in uncovering and preventing alcohol abuse [44,45,46]. However, rather than considering public health interventions targeted at specific subpopulations, our study revealed the need for increased efforts in health promotion aimed at the entire young adult population, knowing that SGM can benefit to a relatively greater extent, so as to narrow health inequalities through a policy that is as proactive and inclusive as possible.

## Figures and Tables

**Table 1 ijerph-18-11724-t001:** Socio-demographic characteristics of the study population.

Demographics	*n* = 9988	%
Age (mean; SD)	20.55	1.81
Non-Italian citizenship	259	2.61%
Degree course		
Health-care professions	2239	23.08%
Sciences	1899	19.57%
Humanities	1575	16.23%
Engineering	1472	15.17%
Law and Economics	607	6.26%
Psychology	932	9.61%
Medicine and Dentistry	978	10.08%
Involved in a stable relationship	4364	44.83%
**Gender identity and sexual orientation**		
Cisgender	9835	98.47%
Female	5932	60.32%
Heterosexual	5606	94.50%
Lesbian/WSW	76	1.28%
Bisexual	250	4.21%
Male	3903	39.68%
Heterosexual	3723	95.39%
Gay/MSM	101	2.59%
Bisexual	79	2.02%
Transgender	12	0.12%
Male	8	0.08%
Female	4	0.04%
Non-disclosed	141	1.41%

**Table 2 ijerph-18-11724-t002:** Prevalence of risk behaviours in the study population.

		Cisgender Females							Cisgender Males						
		Heterosexual		Lesbian/WSW		Bisexual			Heterosexual		Gay/MSM		Bisexual		
		n *	%	n	%	n	%	*p (χ^2^)*	n	%	n	%	n	%	*p (χ^2^)*
Alcohol consumption	≥6 days a week	71	1.3%	0	0.0%	13	5.4%	<0.001	180	5.1%	5	5.2%	6	7.9%	0.440
	2–5 days a week	752	14.1%	8	11.6%	55	22.9%		904	25.6%	23	23.7%	16	21.1%	
	≤1 day a week	3383	63.4%	47	68.1%	142	59.2%		1967	55.7%	49	50.5%	44	57.9%	
	Never	1130	21.2%	14	20.3%	30	12.5%		478	13.5%	20	20.6%	10	13.2%	
Smoking	Current smoker	1098	20.1%	23	31.1%	89	36.2%	<0.001	804	22.2%	25	25.5%	22	28.2%	0.550
	Former smoker	329	6.0%	7	9.5%	38	15.4%		297	8.2%	9	9.2%	8	10.3%	
	Non-smoker	4047	73.9%	44	59.5%	119	48.4%		2514	69.5%	64	65.3%	48	61.5%	
Number of sexual partners within past 24 months	0	394	8.9%	3	5.4%	12	5.7%	<0.001	327	11.6%	7	8.0%	11	16.2%	<0.001
	1	2885	65.1%	42	75.0%	91	43.5%		1496	53.1%	24	27.6%	22	32.4%	
	2	658	14.8%	4	7.1%	51	24.4%		469	16.6%	13	14.9%	13	19.1%	
	3+	494	11.1%	7	12.5%	55	26.3%		526	18.7%	43	49.4%	22	32.4%	
Sexual onset	14 year or less	250	5.8%	2	3.6%	25	12.0%	0.004	133	4.7%	5	5.9%	2	3.2%	0.001
	15–16 year	1518	35.0%	20	35.7%	80	38.5%		835	29.6%	24	28.2%	20	32.3%	
	17–18 year	1904	43.9%	26	46.4%	83	39.9%		1401	49.7%	29	34.1%	24	38.7%	
	19 year or more	663	15.3%	8	14.3%	20	9.6%		451	16.0%	27	31.8%	16	25.8%	
Use of condoms (or other protective barriers)	Never	1299	27.3%	48	76.2%	72	31.9%	<0.001	488	15.3%	19	21.1%	14	19.4%	0.087
	Sometimes	822	17.3%	3	4.8%	56	24.8%		554	17.4%	7	7.8%	14	19.4%	
	Always	2640	55.5%	12	19.0%	98	43.4%		2142	67.3%	64	71.1%	44	61.1%	

* Column totals may not add to total due to missing data.

**Table 3 ijerph-18-11724-t003:** Multinomial logistic regression. The effect of sexual orientation on the adoption of risk behaviours in the study population.

	Cisgender Females (Ref. Heterosexual)							
	Lesbian				Bisexual			
	AOR	95% CI		*p*	AOR	95% CI		*p*
Frequent drinker	n/a				2.84	1.42	5.67	0.003
Current or former smoker	1.08	0.57	2.04	0.817	1.88	1.36	2.61	<0.001
Multiple sexual partnership	0.57	0.21	1.54	0.266	1.81	1.24	2.66	0.002
Early sexual onset	0.61	0.14	2.61	0.509	1.32	0.79	2.21	0.293
Non regular use of condoms	5.76	2.60	12.73	<0.001	1.54	1.11	2.14	0.009
	**Cisgender Males (Ref. Heterosexual)**							
	**Gay**				**Bisexual**			
	**AOR**	**95% CI**		** *p* **	**AOR**	**95% CI**		** *p* **
Frequent drinker	0.56	0.16	1.89	0.347	1.55	0.62	3.87	0.347
Current or former smoker	0.87	0.52	1.47	0.611	1.24	0.68	2.25	0.484
Multiple sexual partnership	5.52	3.26	9.35	<0.001	1.86	1.00	3.48	0.051
Early sexual onset	0.85	0.29	2.51	0.769	0.57	0.13	2.48	0.452
Non regular use of condoms	0.65	0.38	1.14	0.136	1.43	0.79	2.60	0.237

Socio-demographic characteristics not shown; AOR: adjusted odds ratio; Ref.: reference category.

## Data Availability

The datasets used and/or analysed during the current study are available from the corresponding author on reasonable request.

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
