# Peer review of "Sexual and Gender Minorities and Risk Behaviours among University Students in Italy"

_ijerph, 2021, doi:10.3390/ijerph182111724_

Round 1

Reviewer 1 Report

Introduction

  1. The first paragraph of this study is not needed in the introduction. This study focuses on sexual gender minorities (SGM) in Italy not the United States. Start with talking about Italy.
  2. What are the research questions?
  3. Why is this study important? Are there disparities that exist between SGM and heterosexual Italians between the ages of 18 – 25. If so, please talk about it in the introduction.

Methods

  1. Why was there no compensation offered for participation in the study?
  2. Why did the research compare SGM to heterosexual young adults? The authors could have conducted a study with just SGM college students at the two universities.
  3. This is a convenient sample?
  4. What about missing data, how did the authors handle this?
  5. What are the outcomes for the study?
  6. What are the predictors? Were all the predictors single items? Were there measures used in the study? Reliability of these scales?

Reviewer 2 Report

INTRODUCTION:

  • The authors should revisit the use of the term "homosexual" and instead consider "Men who have sex with Men", "Women who have sex with Women", gay or lesbian. According to the NIH Office of Equity, Diversity, and Inclusion, non-LGBTQ+ people should avoid using the term "homosexual". 
  • The introduction lacks background on factors that may contribute to discrimination, stigma, and risk health behaviors.  This should be further developed.
  • Sentence construction and grammar needs to be revisited

MATERIALS AND METHODS:

  • Further description of the categories of variables measured should be included; specifically age, nationality, sexual orientation
  • Information on sexual orientation was collected but there is a lack of mention of gender identity. The authors should consider the differences between sexual orientation and gender identity and the implication for risky health behavior. Further clarity is needed on the distinction of these two variables. 

RESULTS:

  • It is unclear in the results, specifically tables 2 and 3, if when the terms "female" and "male" are used if they refer to both cis and trans or just cis

DISCUSSION

  • Revisit the use of prevalence when referring to SGM, consider  "proportion" instead

CONCLUSION:

  • Consider providing possible recommendations based on study findings

Round 2

Reviewer 1 Report

The authors improved significantly.